# Benefits of Biotics for Cardiovascular Diseases

**DOI:** 10.3390/ijms24076292

**Published:** 2023-03-27

**Authors:** Emília Hijová

**Affiliations:** Center of Clinical and Preclinical Research MEDIPARK, Faculty of Medicine, Pavol Jozef Šafárik University in Košice, Trieda SNP 1, 040 11 Košice, Slovakia; emilia.hijova@upjs.sk

**Keywords:** cardiovascular diseases, cardiometabolic diseases, biotics

## Abstract

Cardiovascular diseases are the main cause of death in many countries, and the better prevention and prediction of these diseases would be of great importance for individuals and society. Nutrition, the gut microbiota, and metabolism have raised much interest in the field of cardiovascular disease research in the search for the main mechanisms that promote cardiovascular diseases. Understanding the interactions between dietary nutrient intake and the gut microbiota-mediated metabolism may provide clinical insight in order to identify individuals at risk of cardiometabolic disease progression, as well as other potential therapeutic targets to mitigate the risk of cardiometabolic disease progression. The development of cardiometabolic diseases can be modulated by specific beneficial metabolites derived from bacteria. Therefore, it is very important to investigate the impact of these metabolites on human health and the possibilities of modulating their production with dietary supplements called biotics.

## 1. Introduction

Cardiovascular diseases (CVD) are a group of heart and blood vessel disorders that include coronary heart disease (CHD, sometimes called ischemic heart disease or coronary artery disease), cerebrovascular disease, rheumatic heart disease, congenital heart disease, and peripheral vascular disease, which affect the blood supply to the heart, brain, and the peripheral regions of the human body [1]. Cardiovascular diseases are the main cause of morbidity and mortality in Europe. The Atlas of Cardiology report, created by the European Society of Cardiology (ESC), provides a current analysis of CVD statistics in 56 member countries. In the European Union (EU), there are more than 6 million new cases of cardiovascular disease and more than 11 million in Europe as a whole each year [2]. Long-term cardiovascular diseases are the most common cause of death in the Slovak population. In 2020, they accounted for 46% of the total number of deaths, mainly caused by coronary heart disease. In men, deaths from cardiovascular diseases accounted for 41% and in women, it accounted for 51.3% [3].

An unhealthy lifestyle, unbalanced diet, alcohol consumption, smoking, and physical inactivity are risk factors for the occurrence and progression of cardiovascular diseases. Among them, in the last two decades, much attention has been paid to the gut microbiota, which has become one of the most innovative areas regarding the study of various diseases. Collectively considered the largest endocrine organ of the body, the gut microbiota, the community of microorganisms that colonizes the gut, play an important role in human health by producing biologically active metabolites that influence many aspects of host physiology. Changes in the structure and function of the microbial community, known as intestinal dysbiosis, are associated with several disease states, including acute or chronic dysfunctions of the host’s cardiovascular system [4,5]. Substances capable of directly affecting the intestine microbiota are called biotics and they include probiotics, prebiotics, synbiotics and postbiotics. The modulation of the microbiota’s composition by biotics seems to be an interesting concept for treating or even preventing the onset of chronic diseases. The aim of this review is to demonstrate how the supplementation of probiotics, prebiotics, synbiotics and postbiotics might be beneficial to the prevention of cardiovascular diseases.

## 2. Cardiovascular Disease, Risk Factors

Cardiovascular diseases, affecting primarily the heart and blood vessels, are currently the leading cause of death worldwide. The World Health Organization reports that cardiovascular diseases are responsible for 17 million deaths annually, accounting for about 31% of all deaths worldwide, and expects this number to increase to more than 23 million by 2030 [6]. Cardiovascular diseases represent a great economic burden for every country. Cardiovascular diseases are complex diseases and include many risk factors that have been reported to contribute to the onset of cardiovascular diseases.

Factors contributing to cardiovascular diseases can be broadly categorized as non-preventable risk factors and preventable risk factors. Age, gender and family history are non-preventable risk factors, and preventable (modifiable, influenceable) risk factors include unhealthy diet, elevated glucose and lipid levels, smoking, physical inactivity, depression and anxiety, stress and gut microbiota. Obesity cause changes in the cardiovascular system by directly affecting the level of adipokines, through inflammation and during the maintenance of vascular homeostasis, or through indirect effects that may be related to a higher prevalence of various diseases, such as hypertension, insulin resistance, hyperglycemia, diabetes mellitus, and dyslipidemia [7].

Therapeutic strategies have been designed and applied based on the variety of risk factors, with drug treatment and lifestyle management being predominant so far [8]. The pharmacologic prevention of cardiovascular diseases using drugs has disadvantages, including an increased end-organ burden, impaired compliance rates, and increased side effects due to drug interactions [9]. These disadvantages have stimulated the development of new treatment options for cardiovascular diseases and underlined the importance of adopting a healthy lifestyle, including consuming functional foods [10]. Modifying the diet is the first line of treatment and offers an effective means of reducing high levels of cholesterol, triacylglycerides and blood sugar, which are the primary risk factors for obesity and are directly associated with the development of cardiovascular diseases and the prevalence of ischemic heart disease in men and women [11]. Functional foods that contain bioactive substances are the key to preventing the progression of cardiovascular diseases by reducing cholesterol and glucose levels, changing the structure and function of the intestinal microbiota and modifying the related immune responses. The imbalance of the gut microbiota affects several metabolic and physiological processes, and changes in this microbial structure are related to the progression of metabolic disorders such as obesity, insulin resistance, atherosclerosis; this imbalance has also been observed in heart failure, thrombosis, atherogenesis, and arterial hypertension [12,13,14].

The positive role of probiotics and prebiotics in changing the microbial and metabolic composition of gut microbiota has been proven, and they can be considered as potential therapeutic and preventive agents for cardiovascular diseases [15]. The therapy of cardiovascular diseases with probiotics and prebiotics can also be explained by the modulation of the host’s immune system. The immunological mechanisms that are promoted by probiotics and prebiotics include changes in dendritic cells, epithelial cells, T regulatory cells, effector lymphocytes, natural killer T cells, and B cells [16]. Many chronic diseases are accompanied by low-grade inflammation, which is also the case with cardiovascular diseases. A systematic review of human studies reported that the abundance of *Faecalibacterium*, *Bifidobacterium*, *Ruminococcus* and *Prevotella* bacteria is, in turn, associated with markers of low-grade inflammation, such as high-sensitivity C-reactive protein (hs-CRP), interleukin (IL) IL-1, IL-6, and tumor necrosis factor alpha (TNF-α). The relationships between the gut microbiota and the markers of low-grade inflammation in humans, and the benefits of a therapeutic strategy for the prevention and treatment of cardiovascular diseases, considering the gut microbiota and its relation to the innate and adaptive immune system, highlight the importance of conducting research on the human gut microbiota as a potential diagnostic tool [17,18,19].

## 3. Gut Microbiota

The human gut tract has approximately 1014 microbes (gut microbiota) and their combined genetic capacities (gut microbiome) have an impact far beyond digestion. The gut microbiota produces the biologically active metabolites that affect the host. While the gut microbiota facilitates many essential and beneficial physiological processes, such as the digestion of macronutrients and the synthesis of certain vitamins, much evidence suggests that the gut microbiota may play a role in the development of adverse phenotypes. Significant changes in the structure and function of the microbial community and their metabolites act as a new risk factor for cardiovascular diseases, and participate in the development and progression of cardiometabolic diseases/disorders (CMD), such as diabetes mellitus, obesity, hypertension, excess body fat and cardiovascular disease [5,20].

The predominant bacteria in a healthy gut microbiota are phyla Firmicutes and Bacteroidetes (90% of the population), followed by Actinobacteria and Proteobacteria; other minor but relevant phyla include Verrucomicrobia and Fusobacteria, although inter-individual variability exists. The diversity, richness, and composition of the gut microbiota vary according to multiple determinants, either endogenous, such as gender, microbial interactions and host genotype [21], or exogenous, such as diet, age, the usage of antibiotics, exercise, smoking, and stress [22].

The gut microbiota is seen as a home for „bioreactors“, which ferment food components and break them down into functional metabolites or microbial products, such as short-chain fatty acids (SCFAs), secondary bile acids, and trimethylamine (TMA). Bacteria-derived metabolites have important functions in the gut during digestion, energy acquisition, gut barrier integrity, and in other organs after they enter the systemic circulation (e.g., glucose circulation in the pancreas, lipid metabolism in the liver, and cognitive functions in the brain). Trimethylamine [23,24], or bile acids [25], are metabolic products that are negatively associated with cardiometabolic diseases, whereas short-chain fatty acids [26], anthocyanin [27] and indoleproprionic acid [28] can positively affect the health of the host. Trimethylamine, an important metabolite, is formed by the human gut microbiota from dietary choline, betaine, and L-carnitine [29]. The microbial conversion of food nutrients that contain the TMA moiety (such as choline, phosphatidylcholine and L-carnitine) in the cholesterol and in fat-rich foods are converted to TMA by specific microbial enzymes, or choline trimethylamine lyase (TMA lyase), via a wide range of metabolic pathways. Mammals do not have TMA lyases, and release part of TMA as a residual product, so that the intestinal microbiota is able to use these nutrients as a source of carbon fuel [30]. After entering the portal circulation of the host, trimethylamine is oxidized by the liver enzyme flavin monooxygenases, especially flavin monooxygenase 3 (FMO3) to trimethylamine N-oxide TMAO, and is excreted by the kidneys. Several articles have summarized the clinical and therapeutic potential of TMAO in cardiometabolic diseases [23,31,32,33]. TMAO is positively correlated with early atherosclerosis, increases the size of the atherosclerotic plaque, promotes the growth of arterial thrombus, and triggers the prothrombotic function of blood platelets [34]. The first study investigating the possible relationship of TMAO levels with insulin sensitivity supports the notion that TMAO levels predict CVD risk independent of insulin sensitivity, which is an important mechanism of cardiometabolic diseases. TMAO levels were positively correlated with age, BMI, fasting blood glucose and blood lipids, and elevated fasting serum TMAO levels were associated with increased carotid intima-media thickness (cIMT), an early marker of atherosclerosis [34]. Atherosclerosis is a chronic inflammatory disease; the increased expression of pro-inflammatory cytokines, including TNF-α and IL-1β, and the decreased expression of the anti-inflammatory cytokine IL-10 have been associated with increased plasma TMAO levels [35]. In addition, TMAO increased the expression and activity of tissue factor through the activation of the nuclear factor kappa B (NF-κB) signaling pathway in primary human coronary artery endothelial cells (HCAECs), thus promoting atherothrombosis [36]. TMAO independently predicted death/myocardial infarction at 2 years but failed to predict death/myocardial infarction at 6 months, and was better than currently used biomarkers [37]. The TMA/FMO3/TMAO pathway, which is controlled by the gut microbiota, is an important regulator of lipid metabolism. The TMAO-generating enzyme FMO3 reduced reverse cholesterol transport, reduced intestinal cholesterol absorption, and altered the composition and size of the bile acid pool [38]. Nutrition is an important factor influencing the levels of TMAO and the progression of atherosclerosis. The results of a clinical and experimental study confirmed the role of the gut microbiota in TMAO metabolism and provided a theoretical basis on the level of the regulation of TMAO by the gut microbiota in the prevention or treatment of atherosclerosis. The use of probiotics, prebiotics and synbiotics, and probiotic functional products is a safer and potentially more effective way to alter the composition of the microbiota c [39,40].

The intestinal microbiota is responsible for the formation of unconjugated free bile acids and secondary bile acids through deconjugation and dihydroxylation reactions. Bile acids can act as signaling molecules involved in inflammation, host metabolism, and may play a role in metabolic disorders and cardiovascular diseases [25,41].

Short-chain fatty acids are produced by anaerobic gut bacteria through the saccharolytic fermentation of resistant carbohydrates (e.g., fructo-oligosaccharides, inulin, resistant starch, sugar alcohols, and polysaccharides from plant cell walls), that escape digestion and absorption in the small intestine. Certain gases, such as hydrogen, methane and carbon dioxide, are produced during fermentation reactions. Although short-chain fatty acids are dependent on diet and the bacteria present in the gut, there are specific foods that contain short-chain fatty acids, such as vinegar, sourdough bread, and dairy products, such as crème fraiche, butter, and cheese. The main short-chain fatty acids formed by gut bacteria are acetate, propionate, and butyrate, which account for approximately 80% of all short-chain fatty acids. Butyrate, propionate, and acetate are present in the colon and feces in an approximately 20:20:60 molar ratio each [42]. However, these levels can vary depending on the microbiota composition, the short-chain fatty acids substrates, and the gut transit time. Acetate can be synthesized by two different pathways [43]. First, acetyl-CoA can be produced by the decarboxylation of pyruvate, then acetyl-CoA is hydrolyzed to acetate by an acetyl-CoA hydrolase. Most acetate is produced by intestinal bacteria, including *Prevotella* spp., *Ruminococcus* spp., *Bifidobacterium* spp., *Bacteroides* spp., *Clostridium* spp., *Streptococcus* spp., *Akkermansia muciniphila*, and *Blautia hydrogenotrophica*, using this pathway. Second, the Wood–Ljungdahl pathway can be used by acetogenic bacteria to produce acetate from acetyl-CoA. Propionate can be synthesized by three different biochemical pathways, namely succinate, acrylate, and propanediol pathway. The succinate pathway is present in only a very limited number of intestinal bacteria, including *Coprococcus catus* [44]. Bacteroidetes and several Firmicutes that belong to the Negativicutes class use the acrylate pathway to form propionate [45]. *Salmonella enterica* serovar Typhimurium and *Roseburia inulinivorans*, just like *Akkermansia muciniphila*, produce propionate by using the propanediol pathway [46]. The butyrate can be synthesized from butyryl-CoA via two different pathways. In the classical pathway, phosphotransbutyrylase and butyrate kinase enzymes are responsible for this conversion [47]. In the second pathway, acetate CoA-transferase converts butyryl-CoA into butyrate and acetyl-CoA using exogenously derived acetate. The latter pathway appears to be preferred by the human gut microbiota rather than the classical pathway, which is restricted to some *Coprococcus* species. *Faecalibacterium prausnitzii*, *Eubacterium rectale*, *Eubacterium hallii*, and *Ruminococcus bromii* present this way and appear to be the major butyrate producers [48]. Short-chain fatty acids act as signaling molecules on intestinal cells, as well as on other tissue cells. This is possible thanks to six receptors to which SCFAs can bind, thus triggering intracellular signaling cascades: free fatty acid receptor 3 (FFAR3 or GPR41), FFAR2 or GPR43, G-protein-coupled receptor 109a (GPR109a or HCAR2), olfactory receptor-78 (Olfr78 in mice or OR51E2 in humans), GPR42 and OR51E1 [49].

Undigested carbohydrates in the gut are fermented by the SCFA-producing bacteria to form acetate, propionate, and butyrate. Short-chain fatty acids can act via two different mechanisms:(1)by directly acting on the enterocytes and maintaining the integrity of the intestinal barrier or(2)by indirectly regulating the inflammatory and immune response, blood pressure, energy intake and use, and lipid and glucose homeostasis, through various mechanisms: (a) the inhibition of lysine/histone deacetylase (K/HDAC), leading to histone hyperacetylation, which causes the higher accessibility of transcription factors to the promoter regions of different genes; (b) the activation of signaling transduction (in the small intestine, colon, liver, spleen, heart, skeletal muscle, neurons, immune cells, and adipose tissues), the secretion of glucagon-like peptide-1 (GLP-1) and peptide YY (PYY) in intestinal enteroendocrine L-cells caused by the binding of SCFAs to the G-protein-coupled receptors, and the increase in cyclic adenosine monophosphate (cAMP) levels by the binding of propionate or acetate to the receptor Olfr78/OR51E2 in vascular smooth muscle cells in the peripheral vasculature and renal afferent arteriole; (c) using butyrate as a ligand of the arylhydrocarbon receptor (AHR) and peroxisome proliferator-activated receptor gamma (PPARγ), leading to the expression of genes dependent on these two transcription factors [50].

Through these indirect and direct mechanisms, short-chain fatty acids may benefit human health, such as by improving the integrity of the gut barrier, by regulating blood pressure and energy intake and consumption, modulating glucose and lipid metabolism, and by mediating the immune system and anti-inflammatory effects (Figure 1).

Anthocyanins are glycosyl-anthocyanidins that are present in high concentrations in vegetables and fruit degraded by gut bacteria to protocatechuic acid (PCA) and free anthocyanidins. The antioxidant, anti-inflammatory and antihyperglycemic properties of PCA may have beneficial effects on cardiovascular diseases and atherosclerosis [27]. Indolepropionic acid (IPA) is a gut microbiota-produced metabolite of dietary tryptophan with the ability to predict the onset of type 2 diabetes mellitus (T2DM). The association between circulating levels of IPA and the confirmed parameters of metabolic syndrome suggests that the composition of the gut microbiome influences IPA levels [28]. Indolepropionic acid is a potential biomarker for the development of T2DM, and may mediate a protective effect by preserving β-cell function and partly by enhancing insulin sensitivity [51].

When the intestinal microbial ecosystem (eubiosis) is balanced, the gut microbiota has important immunological, homeostatic, and metabolic functions that maintain the health of the human host. An imbalance in the gut microbiota, known as dysbiosis, and a reduction in bacterial diversity can lead to metabolic abnormalities, such as inflammation and oxidative stress, which have a negative impact on the physiological states of the host [52]. The composition of the commensal microbiota and its metabolites therefore act as an emerging risk factor for the development of cardiovascular diseases.

## 4. Probiotics and CVD

The influence of gut microbiota includes the consumption of selected beneficial microorganisms, known as probiotics; these are defined as „live microorganisms that, when administered in adequate amounts, confer a health benefit on the host“ [53]. Probiotics have a direct effect on individuals’ health through interactions with the host cells or have an indirect effect by impacting upon other bacterial species. In general, probiotics can adversely affect enteric pathogens via numerous mechanisms:the production of antimicrobial bacteriocins;competitive adhesion to epithelium and mucosa;improving the integrity of the epithelial barrier;and the modulation of the immune system [54,55,56].

Probiotic products may contain one or a combination of several selected microbial strains. Most of them belong to lactic acid bacteria, such as *Lactobacillus*, *Bifidobacterium*, *Lactococcus*, *Streptococcus*, and *Enterococcus*. Yeasts of the genus Saccharomyces are also well-known probiotics [57].

Using probiotics is subject to regulations contained in the general food law, according to which they should be safe for human and animal health. The World Health Organization, the FAO (Food and Agriculture Organization), and EFSA (the European Food Safety Authority) propose that probiotic strains must meet safety and functionality criteria, as well as criteria related to their technological applicability. The safety of a probiotic strain is defined by its origin, its absence of an association with pathogenic cultures and the antibiotic resistance profile. Their functional aspects refer to their survival in the intestinal tract and their immunomodulatory effect. Probiotic strains must meet the requirements associated with their production technology, that is, they must be able to survive and maintain their properties throughout the entire storage and distribution process. Probiotic properties are not associated with the genus or species of the microorganism, but with a few specially selected strains of a certain species [53].

Recent systematic reviews have reported that specific strains of *Lactobacillus* and *Bifidobacterium* are generally used in probiotic therapy, supporting evidence that probiotics may have the potential to reduce cardiovascular disease risk factors, such as obesity [58,59,60] and the blood lipid index, type 2 diabetes mellitus and hypertension [11,61,62,63].

The mechanisms involved in the participation of probiotic bacteria in the hypolipidemic effect can be as follows: the deconjugation of bile salts; the modulation of lipid metabolism; the reduced absorption of intestinal cholesterol through the co-precipitation of intestinal cholesterol with deconjugated bile salts; the incorporation and assimilation of cholesterol into the cell membrane of probiotics; the intestinal conversion of cholesterol to coprostanol; the inhibition of the expression of the intestinal cholesterol transporter Niemann-Pick C1 like 1 (NPC1L1) in the enterocytes; and the inhibition of cholesterol and triglyceride synthesis in the liver by short-chain fatty acids, especially propionic acid [64,65]. Probiotics improve T2DM symptoms, glucose biomarkers and insulin resistance by maintaining the homeostasis of the gut microbiota [66]. A meta-analysis of randomized trials suggests that supplementation with probiotics alone or in a synbiotic combination beneficially effects diabetic patients by improving serum fasting blood glucose (FBG) levels, as well as oxidative stress biomarkers, such as total antioxidant status (TAS), total glutathione (GSH) and malondialdehyde (MDA) [67]. The mechanisms that have been proposed are as follows: improved intestinal integrity, reduced systemic levels of lipopolysaccharide, reduced endoplasmic reticulum stress, reduced stimulation of the proinflammatory genes, including TNF-α, IL-6 and IL-1β, and improved peripheral insulin sensitivity [68,69]. A meta-analysis of 15 randomized control trials with 902 patients focused on the effects of probiotic supplementation on levels of hemoglobin A1c (HbA1c) and FBG; a homeostasis model assessment of insulin resistance (HOMA-IR) in patients with T2DM showed that in the group treated by probiotics, levels of HbA1c (*p* = 0.02), FBG (*p* = 0.003), and HOMA-IR (*p* < 0.00001) were decreased [70].

The effects of probiotics composed of bacterial species are as follows: *Lactobacillus acidophilus*, *Lactobacillus casei*, *Lactobacillus bulgaricus*, *Lactobacillus rhamnosus*, *Bifidobacterium breve*, *Bifidobacterium longum* and *Streptococcus thermophilus*, in combination with fructooligosaccharides, had an effect on fundamental CVD-related parameters, such as the atherogenic indexes of plasma (AIPs), including the ratios of total cholesterol/high-density lipoprotein cholesterol (TC/HDL-C), low-density lipoprotein cholesterol/high-density lipoprotein cholesterol (LDL-C/HDL-C), and log triglycerides/ high-density lipoprotein cholesterol (log TG/HDL-C), systolic and diastolic blood pressure, the Framingham risk score, and antioxidant markers composited from the total antioxidant capacity (TAC), paraoxonase (PON), and the total oxidant status (TOS) in 60 patients with T2DM showed that probiotic supplementation for 6 weeks leads to a significant improvement in the main parameters related to cardiovascular diseases, with the exception of there being no significant changes in antioxidant markers, suggesting the possible beneficial role of probiotics in reducing the risk of future cardiovascular diseases associated with diabetes [71].

The administration of *Lactobacillus reuteri* V3401 to obese adults aged 18 to 65 years who had been experiencing metabolic syndrome for 12 weeks was accompanied by lower levels of inflammation biomarkers, such as TNF-α, IL-6, IL-8, and soluble intercellular adhesion molecule-1, thereby reducing the risk of cardiovascular diseases [72].

Randomized controlled trials (RCTs) published from 1990 to 2020 focused on adults at risk of cardiovascular disease with comorbidities (diabetes, dyslipidemia, metabolic syndrome, hypercholesterolemia, hypertension) and the efficacy of the use of probiotics (yogurt, milk, kefir, powder) on cardiovascular disease risk factors (high blood pressure, overweight body mass index, total cholesterol, LDL-C, HDL-C, elevated HbA1c and fasting glucose), systematically reviewed by Dixon et al. [73]. Probiotics in yoghurt form aided the most significant reductions in the cardiovascular risk factors of total cholesterol, LDL-C, fasting glucose, HbA1c and blood pressure. Probiotics in kefir and powder aided a significant reduction in body mass index (BMI), and elevated HDL-C. One mechanism to explain this characteristic is the greater diversity of probiotic species found in kefir, and their ability to stimulate gastric emptying and shorten absorption and digestion time [74]. Studies with a larger proportion of female patients showed more significant reductions in their outcome, namely in total cholesterol, fasting glucose, HbA1c, LDL-C and BMI. One explanation for this is the relationship between the estrogen–gut microbiome axis, in which women are at a greater risk of microbiota depletion, thus providing a physiological template for the correction of microbiota and a reduction in the risk factors associated with cardiovascular diseases [75].

The targeted management of gut microbiota/metabolome via probiotic administration could be a promising way to protect against coronary artery disease (CAD) [76,77] and arterial hypertension [78,79]. The co-administration of the probiotic strain isolated from the human breast milk of healthy women, *Bifidobacterium lactis* Probio-M8, to patients diagnosed with CAD showed that Probio-M8 synergized with a conventional regimen in order to improve the clinical efficacy of coronary artery disease treatment [80]. The benefits were likely achieved through the probiotic-driven modulation of the gut microbiota and host metabolome, which in turn modulated multiple gut–heart and gut–brain axis pathways, e.g., via increasing the diversity and quantity of anti-inflammatory gut microbial species, reducing serum TMAO/TMA levels, reducing serum IL-6 and LDL-C levels, and modulating specific amino acid and bioactive metabolite levels. Metagenomic analysis showed that significantly more species-level genome bins (SGBs) of *Bifidobacterium adolescentis*, *Bifidobacterium animalis*, *Bifidobacterium bifidum*, and *Butyricicoccus porcorum* were detected in the Probio-M8 group compared to the placebo group, while the abundances of SGBs, represented by *Flavonifractor plautii* and *Parabacteroides johnsonii*, were reduced among the Probio-M8 recipients. The Probio-M8 group achieved better Seattle Angina Questionnaire (SAQ) scores than the placebo group, who received only the conventional regimen, and there was also a reduction in the depression and anxiety levels of the patients. This study highlighted the importance of gut–heart and gut–brain axis regulation in the treatment of coronary artery disease with probiotics.

A clinical study investigated the efficacy and safety of the administration of *Lactobacillus rhamnosus* probiotics in preventing the development of the remodeling process after myocardial infarction (post-MI) in a patient with myocardial infarction (MI) and having undergone successful percutaneous coronary intervention (PCI). Gut dysbiosis has emerged as a new candidate for being associated with the risk of developing heart failure. Selective gut modulation by probiotic administration can improve metabolic dysfunction and attenuate cardiac remodeling (CR) in subjects with myocardial infarction. The potential mechanism of dysbiosis and the onset of cardiac remodeling promotes metabolic endotoxemia, in which gram-negative cell wall lipopolysaccharides (LPSs) are capable of inducing low-grade systemic inflammation, insulin resistance, an increase in the circulation of TMAO, and an increased cardiovascular risk, which may be attenuated by probiotics. Lipopolysaccharides that bind to Toll-like receptor (TLR) 4 induces the stimulation of the innate immune system, leading to a proinflammatory response and subsequent metabolic disturbances. The results of this study provide evidence for changes in the gut microbiota when taking probiotics in order to prevent cardiac remodeling after myocardial infarction [81].

Stroke is the leading cause of morbidity and mortality among the cardiovascular diseases. Approximately 50% of stroke patients are associated with gastrointestinal complications, which include gut dysmotility, leaky gut syndrome, dysbiotic gut microbiota, and sepsis of intestinal origin; these patients have poor outcomes after stroke [82]. The underlying mechanisms of these problems remain poorly understood. One possible explanation is the two-way communication between the gut and the nervous system, the so-called gut–brain axis. Activation of the sympathetic nervous system (SNS) is associated with inflammation-induced vascular endothelial dysfunction and cardiometabolic diseases [83]. Since some gut bacteria, such as the genera *Bifidobacterium* and *Lactobacillus*, can produce neurotransmitters, probiotics supplementation can cause changes in the gut microbiota by altering the types and concentrations of neurotransmitters, which in turn affects the sympathetic nervous system and subsequent cardiometabolic activities [84].

## 5. Prebiotics and CVD

The positive modulation of gut microbiota is the application of substrates/ingredients that can be selectively used by beneficial bacteria and can promote their growth and the production of necessary metabolites. Substrates/ingredients increasing the number of beneficial microbes were designated as prebiotics, and the concept of prebiotics has been accepted. The term prebiotics was defined in 1995 as an indigestible food ingredient that beneficially affects the host by selectively stimulating the growth and/or activity of one or a limited number of bacteria in the colon, and that thus improves the health of the host [85]. The definition of prebiotics was updated in 2004 as “a selectively fermented ingredient, that enables specific changes in composition and/or activity in the gut microbiota that confers benefits to the health of the host” [86]. The authors of the concept evaluated prebiotics according to the following three criteria:resistance to gastric acid and hydrolysis by mammalian enzymes and gastrointestinal absorption;the ability to be metabolized by the gut microbiota;and selective stimulation of growth and/or activity of the gut bacteria for health maintenance.

This definition was widely accepted, and was later updated in order to support, for example, its application in extraintestinal sites. The definition of prebiotics has changed considerably over the past two decades, indicating the need for consensus on the specificity, mechanisms of action, health attributes, and importance of prebiotic substrates. Recently, the definition of a prebiotic has been updated to a substrate that is selectively utilized by host microbiota, resulting in health benefits [87].

There are five basic criteria used to classify food ingredients as potential prebiotics according to Wang [88]:Resistance to digestion in the upper parts of the intestinal tract;Fermentation with potentially beneficial bacteria in the colon;Beneficial effects on the health of the host—fermentation can lead to the production of various short-chain fatty acids, an increase in stool weight, a slight decrease in the pH of the colon, a reduction in nitrogenous end products and fecal enzymes, and an improvement in the immune system;The selective stimulation of the growth of probiotics.Stability in different food/feed processing conditions—prebiotics must be able to withstand food processing conditions and remain chemically unchanged, undegraded and available for bacteria in the gut.

Resistant oligosaccharides fructans [fructooligosaccharides, oligofructose, and inulin] and galactans [galactooligosaccharides (GOS), and trans-galactooligosaccharides (TOS)] areclassified as prebiotics by definition and are those most commonly used. The definition of prebiotics has been expanded to include human milk oligosaccharides (HMOs) and non-saccharide substances, such as polyphenols and polyunsaturated fatty acids (PUFA), as potential new classes of prebiotics. In addition, prebiotics should not have adverse effects on the host, such as the growth of pathogenic microorganisms or abdominal distention caused by excessive gas production [87,89]. Some types of fiber can be classified as prebiotics. Dietary fiber is made up of carbohydrate polymers with three or more monomeric units (MU), which are neither digested nor absorbed in the human intestine. Dietary fiber includes the following: (1) non-starch polysaccharides (NSP) from fruits, vegetables, and own or extracted cereals and tubers, which are chemically, physically and/or enzymatically modified or synthetic (MU ≥ 10); (2) resistant (non-digestible) oligosaccharides (RO) with (MU 3–9), such as galactosides, galactans, fructans, xylooligosaccharides, mannanoligosaccharides; and (3) resistant starch (RS) with (MU ≥ 10), and (4) associated substances (non-carbohydrates), which are linked with the polysaccharide present in the cell wall of plants [90].

Fiber consumption has health benefits that include improving intestinal function through the composition and metabolic activity of gut bacterial communities, the production of short-chain fatty acids, by lowering blood glucose and cholesterol levels, increasing satiety, and reducing the risk of chronic diseases [91,92]. Prebiotic dietary fibers act as a selective substrate for beneficial bacteria in the gut, such as *Bifidobacterium* and *Lactobacillus*, which are beneficial to the health of the host [89] and play a key role in shaping this microbial ecosystem. It has been suggested that high-fiber diets, such as Mediterranean and vegetarian diets, may protect against the development of cardiometabolic diseases, such as blood pressure, cholesterol, body weight, and systematic inflammation, including cardiovascular diseases, obesity, and type 2 diabetes mellitus, by modulating the gut microbiota. Dietary fiber and prebiotics, as many of these polysaccharides are metabolized by the gut microbiota, lead to the production of short-chain fatty acids. These metabolites of prebiotic fermentation exhibit anti-inflammatory and immunomodulatory capabilities [93,94]. The gut microbiota benefits humans through the production of short-chain fatty acids via carbohydrate fermentation, and a lack of short-chain fatty acid production is associated with type 2 diabetes mellitus. In a randomized clinical trial, it was shown that a select group of SCFA-producing strains was promoted by dietary fibers and that most other potential producers were either reduced or unchanged in T2DM patients. When fiber-supported SCFA producers were present with greater diversity and abundance, particularly acetate and butyrate, participants experienced improvements in HbA1c levels, in part through the increased production of glucagon-like peptide-1 [95]. A clinical study investigated the effect of a dietary portfolio (DP) based on a combination of functional foods, including inulin, chia seeds, soy protein, dehydrated nopal, omega-3 fatty acids, vegetable protein, polyphenols, and soluble and insoluble fiber in patients with T2DM. Patients with T2DM showed intestinal dysbiosis, which was characterized by an increased prevalence of *Prevotella copri*. A dietary intervention with functional foods significantly changed the fecal microbiota compared to the placebo, by increasing the alpha diversity and modifying the abundance of specific bacteria, independent of antidiabetic drugs. There was a decrease in *Prevotella copri* and an increase in *Faecalibacterium prausnitzii* and *Akkermansia muciniphila. Faecalibacterium prausnitzii* and *Akkermansia muciniphila* are bacterial species that promote SCFA production and are known for their anti-inflammatory effects. The group with a diet containing functional foods high in fiber, polyphenols, and plant proteins experienced a significant reduction in glucose, total cholesterol and LDL-C, free fatty acids (FFAs), triglycerides, HbA1c, lipopolysaccharides, C-reactive protein (CRP), and an increase in antioxidant activity. This diet provides benefits for the composition of the fecal microbiota and may offer potential therapies with the aim to improve glycemic control, dyslipidemia, and inflammation [96]. A vegetarian diet (lacto-ovo-vegetarian diet allowing the intake of eggs and dairy products), in conjunction with optimal medical therapy, can reduce cardiovascular therapy, decrease the levels of oxidized LDL-C, improve cardiometabolic risk factors (lipids, HbA1c, hs-CRP, BMI, blood pressure, heart rate, quality of life, SCFA, TMAO, plasma metabolome), and change the relative abundance of intestinal microbiota genera, such as those within the families of *Ruminococcaceae*, *Lachnospiraceae*, and *Akkermansiaceae* and in plasma metabolites (L-carnitine, acylcarnitine, phospholipids), in patients with ischemic heart disease [97]. *Ruminococcaceae* were predominant gut bacteria and were associated with the clinical symptoms and hemodynamics of vasovagal syncope (VVS), suggesting that gut microbiota could be involved in the development of VVS in children [98]. No work has been published in which probiotics or prebiotics are applied in VVS. The recognition of gut microbes as participants in the pathogenesis of cardiovascular and metabolic diseases suggests that pharmacological interventions aimed at “influencing the microbiome” may potentially serve as a therapeutic approach in the treatment of cardiometabolic diseases. The proof of concept for the idea that the inhibition of microbial TMA production through microbial TMA lyase inhibition may serve as a potential therapeutic approach to the prevention or treatment of atherosclerosis is provided in this study. DMB (3,3-dimethyl-1-butanol), a structural analog of choline, and a non-lethal inhibitor of microbial enzymes that target causal pathways, once identified, may serve as an effective and complementary strategy to the approaches used currently in order to prevent and treat cardiometabolic diseases [99].

## 6. Synbiotics and CVD

The supply and use of high doses of commercial probiotics without the participation of dietary prebiotics would not be useful and would not produce the secondary metabolites that have beneficial effects on the host; therefore, it is not surprising that synbiotics have emerged as a substitute, in order to offer additional benefits compared to the use of probiotics or prebiotics alone (Figure 2).

Probiotics are effective in the small and large intestine, and the effects of prebiotics are mainly observed in the large intestine. The combination of these two substances can have synergistic results, the main goal being to increase the survival of probiotic microorganisms in the gastrointestinal tract. The International Scientific Association for Probiotics and Prebiotics (ISAPP) has updated the definition of synbiotics to “a mixture containing live microorganisms and substrate(s) selectively utilized by host microorganisms that provide a health benefit to the host”.

The following two types of synbiotics have been described:(1)synergistic synbiotics—synbiotics in which the substrate is proposed to be used selectively and is co-administered to microorganism(s).(2)complementary synbiotics—synbiotics composed of a probiotic in combination with a prebiotic that is intended to target indigenous microorganisms [100]. Two modes of synbiotic action are known: a, action through the improved viability of probiotic microorganisms, and b, action through the provision of specific health effects.

The synbiotics used and a nutritional diet are associated with positive changes in the number of selected genera in the intestinal bacteria that form the intestinal microbiota, and also with a favorable effect on cardiometabolic risk factors (overweight and obesity, lipid profile, inflammation, glucose level, blood pressure). Obese and overweight adults after 6 months of taking *Bifidobacterium animalis* in combination with Litesse Ultra polydextrose had a reduced weight, a reduced plasma bile acid level, altered gut microbiota, increased accounts of *Akkermansia*, *Christensenellaceae* and *Methanobrevibacter*, an improved gut barrier function, and improved obesity-related markers [101].

Healthy overweight volunteers after a 3-month application of *Lactobacillus acidophilus*, *Bifidobacterium lactis*, *Bifidobacterium longum*, *Bifidobacterium bifidum* and a mixture of trans-galactooligosaccharide had a reduced BMI, waist circumference, body fat mass and HbA1c, and increased amounts of *Bifidobacterium*, *Lactobacillus*, *Ruminococcus*, and *Verrucomicrobiae* [102].

In subjects with metabolic syndrome, the intake of synbiotics (*Lactobacillus casei*, *Lactobacillus rhamnosus*, *Streptococcus thermophilus*, *Bifidobacterium breve*, *Lactobacillus acidophilus*, *Bifidobacterium longum*, *Lactobacillus bulgaricus* and fructooligosaccharides) reduced their BMI, fasting blood glucose, HOMA-IR, and increased their levels of GLP-1 and peptide YY (PYY) [103].

A randomized placebo-controlled clinical trial in elderly patients with metabolic syndrome indicated that supplementation with *L. plantarum*, *L. acidophilus*, *L. reuteri*, in combination with inulin and fructooligosaccharide, improved their visceral adiposity, reduced their waist circumference, lowered their lipid profile (TC, HDL-C, triglycerides), hs-CRP and TNF-α, and reduced their metabolic syndrome and blood pressure [104]. A commercially available synbiotic (Sanprobi Super Formula^®^, Poland), containing seven live strains of probiotic bacteria (*Bifidobacterium lactis* W51, *Bifidobacterium lactis* W52, *Lactobacillus acidophilus*, *Lactobacillus plantarum*, *Lactobacillus paracasei*, *Lactobacillus salivarius*, *Lactobacillus lactis*) and two prebiotics (fructooligosaccharide, inulin), was administered in a dose of 2–4 capsules per day to subjects with BMI ≥ 25 kg/m^2^. A significant increase in the number of *Lactobacillus* spp. and H_2_O_2_
*Lactobacillus* and a decrease in the number of proteolytic bacteria (*Escherichia coli Biovare*, *Proteus* spp., *Pseudomonas* spp.) were noted in the synbiotic group after 12 weeks of the diet. A biomarker of impaired intestinal barrier function, zonulin, was significantly reduced in the synbiotic group [105]. The consumption of yogurt that was enriched with synbiots (*Streptococcus thermophilus*, *Lactobacillus delbrueckii subsp. bulgaricus*, *Bifidobacterium animalis subsp. lactis* BB-12 and whey protein, inulin, calcium, and vitamin D3) for ten weeks significantly reduced body fat mass and improved body parameters, such as blood pressure, insulin sensitivity, and lipid profiles in obese patients with metabolic syndrome [106]. Two articles report that supplementation with synbiotics, particularly containing probiotic strains (*Lactobacillus acidophilus*, *Lactobacillus casei*, *Bifidobacterium bifidum*) and prebiotic “inulin”, in patients with coronary heart disease produces an improvement in the levels of markers of insulin metabolism, HDL-C, a reduction in serum hs-CRP and MDA, and elevated plasma nitric oxide [107,108].

## 7. Postbiotics and CVD

Clinical specialists are increasingly recognizing the link between the gut microbiota and the etiology of obesity and cardiometabolic disorders. Gut microbiota can affect the cardiometabolic phenotype by fermenting indigestible food components and thus produce various bioactive metabolites.

Although the exact definition of postbiotics is debated, postbiotics include any substance released or produced by the metabolic activity of the microbiota that has beneficial effects on the host, directly or indirectly [109,110]. Postbiotics, although not containing live bacteria, present benefits to host health through mechanisms similar to those involved in probiotics. The currently available postbiotics include the following:

(1) cell-free supernatants with biologically active metabolites that are secreted by bacteria and yeast into the surrounding liquid; (2) exopolysaccharides produced by biopolymers, released from the microbiota; (3) antioxidant enzymes that are generated by the microbiota in order to protect against reactive oxygen species; (4) cell wall fragments, including bacterial lipoteichoic acid (LTA); (5) SCFAs, metabolites involved in the breakdown of plant polysaccharides by the gut microbiota; (6) bacterial lysates obtained via the degradation of Gram-positive and Gram-negative bacteria; and (7) metabolites produced by intestinal microbiota [111].

Short-chain fatty acids as metabolites are probably the most studied postbiotics derived from gut bacteria and have been proposed as potential disease-modifying and/or disease-preventing factors in cardiometabolic diseases, including type 2 diabetes mellitus, obesity, and cardiovascular diseases, among others. The most abundant short-chain fatty acid, acetate, favorably affects host energy and substrate metabolism through the secretion of GLP-1 and PYY gut hormones, thus affecting appetite, reducing whole-body lipolysis, lowering systemic levels of pro-inflammatory cytokines, and increasing energy expenditure and fat oxidation [112].

Short-chain fatty acids are not only important for gut health as signalling molecules, but can also enter the systemic circulation and directly affect the metabolism or function of peripheral tissues. They can favorably modulate adipose tissue, energy metabolism, skeletal muscle and liver tissue, and contribute to improvements in glucose homeostasis and insulin sensitivity [113]. Therefore, well-controlled human intervention studies that investigate the effect of short-chain fatty acids on cardiometabolic health are eagerly awaited. Postbiotics, such as MDP (muramyl dipeptide) and Amuc_1100 (component of the cell wall of bacterium *Akkermansia municiphila*, known as a next generation probiotic-NGP), are promising candidates for promoting cardiometabolic benefits through innate immune activation [114,115]. Other bacteria-derived molecules, such as certain LPSs and TMAO, are pathognomonic for immunometabolic mechanisms that exacerbate the risk of cardiovascular disease. TMAO is well-positioned as a biomarker and a possible therapeutic target to alleviate cardiometabolic complications. The daily consumption of probiotic strains *Lactobacillus acidophilus*, *Lactobacillus rhamnosus* GG, *Bifdobacterium animalis*, *Bifidobacterium longum* decreased serum TMAO levels in healthy adult males [116].

Although the potential mechanisms involved in the effect of postbiotics on cardiovascular diseases have not been elucidated, it would be of great interest to investigate the beneficial role of postbiotics in these diseases.

## 8. Conclusions

A growing number of scientific reports indicate that the microbiota and the maintenance of homeostasis in the intestinal environment are factors that contribute to the health of the host. Different strategies are required to maintain intestinal homeostasis/eubiosis. In this context, the inclusion of biotics (probiotics, prebiotics, and synbiotics) in the diet as nutritional supplements appears to be a promising and effective approach to correcting the gut dysbiosis and the metabolic functions of the host. Gut dysbiosis, a disturbed intestinal barrier, and persistent low-grade inflammation represent a “vicious circle” in the etiopathology of metabolic diseases. Well-designed clinical trials are needed in order to clearly demonstrate the role of biotics in restoring the intestinal environment in patients with metabolic disorders. The production of new functional foods enriched with promising next-generation probiotics, in combination with prebiotics, represents a new targeted strategy that is currently being developed in order to reshape the gut microbiota and reverse metabolic changes in the organism, which can help reduce the risk factors that cause cardiometabolic diseases.

## Figures and Tables

**Figure 1 ijms-24-06292-f001:**
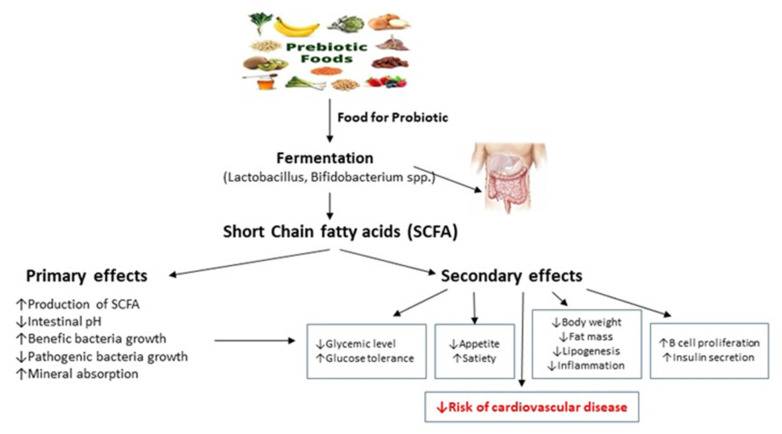
Primary and secondary effects of short-chain fatty acids. More details in text. Arrows indicate the action of prebiotics and probiotics on the production of SCFA. Primary and secondary effects of SCFA leads to a reduction in the risk of cardiovascular disease.

**Figure 2 ijms-24-06292-f002:**
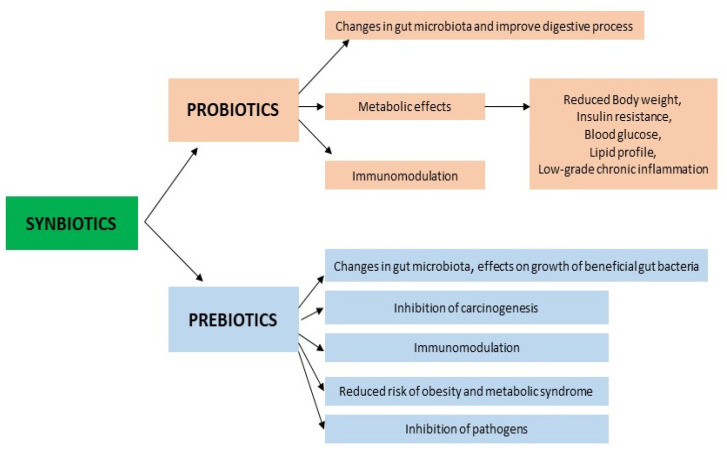
Mechanisms and effects of synbiotics. More details in text.

## Data Availability

Not applicable.

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
