# Peer review of "Benefits of Biotics for Cardiovascular Diseases"

_ijms, 2023, doi:10.3390/ijms24076292_

Round 1
Reviewer 1 Report
The authors explained the role of biotics and CVD in detail and also summarized the importance of biotics in preventing the risk of CVD along with drug consumption. The manuscript has some major issues.
Comments:
1. Similarity index was more than 70% in the plagiarism check.
2. Page 2, line 65 has the spelling mistake found "Owerveight" instead of "overweight".
3. Page 3, the following sentence needs citation "Probiotics and prebiotics which play a positive roles in altering the microbial and metabolic composition of intestinal microbiota could be considered as a potential therapeutic and preventive agents for CVD".
4. Page 4, the following sentence needs citation "TMAO is positively correlated with early atherosclerosis, increases atherosclerotic plaque size, triggers prothrombotic platelet function, and promotes arterial thrombus growth".
5. Page 5, the following sentence is misleading "In the colon and stool, butyrate, propionate, and acetate are found in an approximate molar ratio of 20:20:60, respectively, although these values vary depending on the microbiota composition, SCFA substrates, and gut transit time [40]", change the sentence (for example "Butyrate, propionate, and acetate are present in the colon and feces in approximately 20:20:60 molar ratio each [40]. However, these levels can vary depending on the microbiota composition, SCFA substrates, and gut transit time).
6. On Page 7, line 262 the sentence "idirectly through effects on other bacterial species" has a grammatical mistake so change the sentence correctly "indirectly through effects on other bacterial species".
7. On page 8, lines 311 - 320 sentences were difficult to understand, So reframe the sentence in a more convenient manner. In addition, while describing bacteria the flow was interrupted by adding "fructooligosaccharides" which was misleading. Also, reference was missing in the content mentioned.
8. Page 9, the following sentence "Probiotics in yoghurt form demonstrated the most significant reductions in the cardiovascular risk factors of total cholesterol, fasting glucose, HbA1c, LDL-C; and blood pressure", add the citation.
9. Page 12, the sentence in 480-487 lines needs clarity.
10. Page 13, line 521 "use of synbiotics aree associated", what it means.
11. Recent article (PMID: 36771313) explained the role of microbiome and inflammation in CVD, So including this article will add value to the manuscript.
12. The whole manuscript needs an extensive grammar check (e.g. misuse of punctuation, adjunction, spelling mistakes, and so on).
13. The expansion of the acronym was repeatedly used more than once (e.g. CVD, SCFA, and so on).
Author Response
Thank you for the review and I attach explanations, proposed necessary misspelling have been repaired in the manuscript.
The authors explained the role of biotics and CVD in detail and also summarized the importance of biotics in preventing the risk of CVD along with drug consumption. The manuscript has some major issues.
Comments:
- Similarity index was more than 70% in the plagiarism check.
I will try to reduce the similarity index.
- Page 2, line 65 has the spelling mistake found "Owerveight" instead of "overweight".
It's a grammatical error, overweight is correct.
- Page 3, the following sentence needs citation "Probiotics and prebiotics which play a positive roles in altering the microbial and metabolic composition of intestinal microbiota could be considered as a potential therapeutic and preventive agents for CVD".
Suplemented reference
Markowiak, P.; Åšlizewska, K. Effects of Probiotics, Prebiotics, and Synbiotics on Human Health. Nutrients 2017,9,1021. doi: 10.3390/nu9091021.
- Page 4, the following sentence needs citation "TMAO is positively correlated with early atherosclerosis, increases atherosclerotic plaque size, triggers prothrombotic platelet function, and promotes arterial thrombus growth".
This sentence corresponds to citation 32, the text continues.
- Page 5, the following sentence is misleading "In the colon and stool, butyrate, propionate, and acetate are found in an approximate molar ratio of 20:20:60, respectively, although these values vary depending on the microbiota composition, SCFA substrates, and gut transit time [40]", change the sentence (for example "Butyrate, propionate, and acetate are present in the colon and feces in approximately 20:20:60 molar ratio each [40]. However, these levels can vary depending on the microbiota composition, SCFA substrates, and gut transit time).
The recommended sentence change is included in the text.
- On Page 7, line 262 the sentence "idirectly through effects on other bacterial species" has a grammatical mistake so change the sentence correctly "indirectly through effects on other bacterial species".
The recommended sentence changes is included in the text.
- On page 8, lines 311 - 320 sentences were difficult to understand, So reframe the sentence in a more convenient manner. In addition, while describing bacteria the flow was interrupted by adding "fructooligosaccharides" which was misleading. Also, reference was missing in the content mentioned.
Lines 311-324 belong to the references Ahmadian, F. et al., 2022
- Page 9, the following sentence "Probiotics in yoghurt form demonstrated the most significant reductions in the cardiovascular risk factors of total cholesterol, fasting glucose, HbA1c, LDL-C; and blood pressure", add the citation.
Citation was added.
- Page 12, the sentence in 480-487 lines needs clarity.
The author describes the effect of a vegetarian diet on cardiometabolic risk factor.
- Page 13, line 521 "use of synbiotics aree associated", what it means.
It´s a grammatical error, synbiotics are associated is correct
- Recent article (PMID: 36771313) explained the role of microbiome and inflammation in CVD, So including this article will add value to the manuscript.
Recommended article has been included to the manuscript.
- The whole manuscript needs an extensive grammar check (e.g. misuse of punctuation, adjunction, spelling mistakes, and so on).
I accepted this opinion.
- The expansion of the acronym was repeatedly used more than once (e.g. CVD, SCFA, and so on).
I accepted this opinion.
Reviewer 2 Report
The review presented for review accumulates the results of numerous experimental and clinical studies devoted to the study of the role of microbiota in the development of cardiovascular, cerebrovascular and metabolic diseases. Currently, the microbiota is considered as a new possible target in the prevention and treatment of cardiovascular diseases. The microbiota is interrelated with the factors underlying atherogenesis, increased vascular wall stiffness, the development of cardiovascular diseases, and the processes of inflammation. The study of the relationship between the state of the intestinal microbiota and CVD risk factors opens up new opportunities for the development of individual programs for the prevention and treatment of CVD.
The manuscript submitted for review is clear, presented in a well-structured form. The cited references are relevant.
The data analyzed by the authors may be of considerable interest both for a professional audience regarding the development of CVD prevention programs and for a wide range of readers of the journal.
Nevertheless, in the process of reviewing the submitted manuscript, I had some questions:
1. In my opinion, the wording of the title of the article requires correction, since the authors present data on the positive role of changes in the microbial and metabolic composition of the intestinal microbiota, which can be considered as potential therapeutic and preventive agents for both cardiovascular and metabolic diseases.
2. The phrase in line 53: "This section may be divided by subheadings. It should provide a concise and precise..." - perhaps this is a note?
3. You need to check the following links:
«3. National Health Information Center, Bratislava 2021, ISBN 978-80-89292-80-6, 260 p.»;
«31. Koeth, R.A.; Wang Z, Levison BS, Buffa JA, Org E, Sheehy BT, Britt EB, Fu X, Wu Y, Li L, Smith JD, DiDonato JA, Chen J, Li H, Wu 705 GD, Lewis JD, Warrier M, Brown JM, Krauss RM, Tang WHW, Bushman FD, Lusis AJ, Hazen SL. Intestinal microbiota metabolism 706 of L-carnitine, a nutrient in red meat, promotes atherosclerosis. Nat. Med. 2013,19, 576–585. doi: 10.1038/nm.3145».
Author Response
Thank you for the review and I attach explanations, proposed necessary misspelling have been repaired in the manuscript.
Nevertheless, in the process of reviewing the submitted manuscript, I had some questions:
- In my opinion, the wording of the title of the article requires correction, since the authors present data on the positive role of changes in the microbial and metabolic composition of the intestinal microbiota, which can be considered as potential therapeutic and preventive agents for both cardiovascular and metabolic diseases.
In my opinion, the title corresponds to the content of the manuscript, which states predominantly positive effects of changes in the microbial and metabolic composition, while most of the cited authors consider supplementing this data with new clinical evidence.
- The phrase in line 53: "This section may be divided by subheadings. It should provide a concise and precise..." - perhaps this is a note?
Yes, it is my note.
- You need to check the following links:
«3. National Health Information Center, Bratislava 2021, ISBN 978-80-89292-80-6, 260 p.»;
It is correct, National Health Information Center in Slovak republic, added Slovak republic.
«31. Koeth, R.A.; Wang Z, Levison BS, Buffa JA, Org E, Sheehy BT, Britt EB, Fu X, Wu Y, Li L, Smith JD, DiDonato JA, Chen J, Li H, Wu 705 GD, Lewis JD, Warrier M, Brown JM, Krauss RM, Tang WHW, Bushman FD, Lusis AJ, Hazen SL. Intestinal microbiota metabolism 706 of L-carnitine, a nutrient in red meat, promotes atherosclerosis. Nat. Med. 2013,19, 576–585. doi: 10.1038/nm.3145».
Reference is correct, 705 is line number.

Round 2
Reviewer 1 Report
Dear Author,
Dear author,
Thanks for implementing the suggested changes in the revised version. However, I still noticed plagiarism throughout the manuscript. Henceforth, manuscript required extensive paraphrasing before acceptance.
Author Response
Cover letter for reviewer 1
Thank you for the review and I attach explanations, proposed necessary misspelling have been repaired in the manuscript.
Reviewer:
Dear author,
Thanks for implementing the suggested changes in the revised version. However, I still noticed plagiarism throughout the manuscript. Henceforth, manuscript required extensive paraphrasing before acceptance.
Author:
Dear reviewer
Paraphrasing is a modified idea of another author, whose work is cited in new manuscript, while the meaning of the idea remains unchanged. Information from other authors' works is freely paraphrased in my work, therefore it seems that the work is plagiarized.
I checked the whole manuscript and made changes.
